# Improving the Accuracy of Predicting Bank Depositor's Behavior Using a Decision Tree

Fereshteh Safarkhani [1] and Sérgio Moro [2,*]

1    Department of Computer Engineering, Islamic Azad University Islamshahr Branch, Teheran 33147-67653, Iran; Fereshteh.safarkhani@gmail.com
2    Instituto Universitário de Lisboa (ISCTE-IUL), ISTAR, 1649-026 Lisboa, Portugal
*    Correspondence: Scmoro@gmail.com

**Abstract:** Telemarketing is a widely adopted direct marketing technique in banks. Since customers hardly respond positively, data prediction models can help in selecting the most likely prospective customers. We aim to develop a classifier accuracy to predict which customer will subscribe to a long-term deposit proposed by a bank. Accordingly, this paper focuses on a combination of resampling, in order to reduce the imbalanced data, using feature selection, to reduce the complexity of data computing and dimension reduction of inefficiency data modeling. The performed operation has shown an improvement in the performance of the classification algorithm in terms of accuracy. The experimental results were run on a real bank dataset and the J48 decision tree achieved 94.39% accuracy prediction, with 0.975 sensitivity and 0.709 specificity, showing better results when compared to other approaches reported in the existing literature, such as logistic regression (91.79 accuracy; 0.975 sensitivity; 0.495 specificity) and Naive Bayes classifier (90.82% accuracy; 0.961 sensitivity; 0.507 specificity). Furthermore, our resampling and feature selection approach resulted in improved accuracy (94.39%) when compared to a state-of-the-art approach based on a fuzzy algorithm (92.89%).

**Keywords:** machine learning; data mining; Artificial Intelligence

## 1. Introduction

Data mining is key to extracting insightful knowledge about a problem to which large sets of information exist. It offers a methodological prescriptive approach to deal with the information explosion in current industries so as to obtain valuable information [1]. Data mining can be used to support the decision-making process in order to achieve goals defined by companies; one of the most common data mining tasks adopted in marketing is classification [2]. Data-driven decision-making methods have an important role in responding to the harsh business environment [3]. Specifically, customer targeting consists of identifying prospective buyers of a product (or subscribers of a service) offered within the context of a marketing campaign [4]. By incorporating classification models (i.e., classifiers) into decision support systems, decision makers can determine the most beneficial data mining models in order to improve the business [5]. The large number of available datasets in companies makes data mining a useful approach for finding the information that managers may use to make better decisions. Additionally, knowledge discovery in databases processes (KDD) can help to obtain better models of data [6]. KDD and data mining are not the same. KDD points to the whole process of knowing the valuable data information. Data mining points to identifying new models by focusing on models that obtain valuable information from the database [6].

Nowadays, banks invest a lot in marketing, and related research has emerged to become mainstream in academic literature. Some methods for automatic classifiers with non-statistical classification were investigated in [7], such as the alternating decision tree (AD Tree), random forest, best-first decision tree (BF Tree), C4.5 (J48), RBF (radial basis function), MLP (multilayer perceptron) and SVM (support vector machine); the study

achieved an accuracy of 90.20% using the J48 in a process that took 10 s. Ref. [8] compared four data mining models: J48 graft, radial basis function network, SVM, and logical analysis of data (LAD tree). The best result was obtained using the J48, with an accuracy rate of 76.52%, and the SVM (an advanced classification algorithm) in classification and regression processes. Ref. [9] achieved the highest sensitivity, specificity and accuracy of 86.95% in a bank dataset. Ref. [10] focused on feature selection to reduce the extra features from bank marketing datasets. Ref. [11] achieved an accuracy of 91.19%. After removing extra features and with just 3 features out of 20, the same authors were able to increase the accuracy for predicting the bank long-term deposit subscription within telemarketing campaigns. Ref. [5] proposed an approach to predict the telemarketing success in the bank industry for selling long-term deposits. Altogether, four data mining models were compared: neural networks (NN), logistic regression (LR), decision trees (DT) and support vector machines (SVM); the best results were obtained by the NN. Ref. [12] stated that by anticipating the bank's potential customers who wanted to open long-term deposits reduced marketing costs by saving wasted campaign resources. Ref. [13] used different trained neural networks to reduce the possible risk of unpleasant noise. Other recent studies tended to also use black-box models such as NN and SVM, such as [3].

This paper focuses on data mining classification and the performance of the most important decision tree algorithm called J48. To evaluate the result, models were compared on accuracy, sensitivity and specificity in the bank marketing dataset [11]. The aim of this study is to predict the successful selling of long-term deposits to customers by using bank marketing. Additionally, we adopted resampling to deal with the imbalanced challenge of such a dataset.

Details of the bank dataset, preprocessing, feature selection, data mining techniques of the paper are provided in Section 2. Section 3 includes the evaluation metrics for the trained and tested models. In Section 4, results achieved using the J48 model are shown and discussed. Conclusions are drawn in Section 5.

## 2. Materials and Methods

This study uses popular methodology of data mining called the cross-industry standard process for data mining known as CRISP-DM. CRISP-DM is a proven method to guide the efforts of the mining industry. This methodology contains six steps to make the data mining model useful for decision making [14]. The first step emphasizes understanding the project objectives and requirements and converts them into definitions for the field of data mining. After that, there is original data collection and activities that identify data quality problems, discover the nature of the data and find interesting subsets from hidden information. The next step is data preparation and includes all activities that construct the final dataset out of the raw data.

Data preparation tasks are usually performed several times. This includes selecting tables, features and transformations with data modeling tools. In the modeling phase, various modeling techniques are selected, applied, and adjusted so that the model is optimized to obtain the results properly. Usually, the various techniques are used for solving data mining.

Before going to the assessment phase, the models should be assessed. The key objective of the assessment phase is to investigate the issues that have not been sufficiently considered. At the end of this phase, a decision is taken regarding the application of the obtained data mining results. In the final phase of data mining, if the objective resulted in increased knowledge of the issues and the customer is organized and present, the knowledge gained should be used. Depending on the requirements, the deployment phase can be created simply or complexly through implementation as a simple report and repeatable data mining process [14]. The design approach is shown in Figure 1.

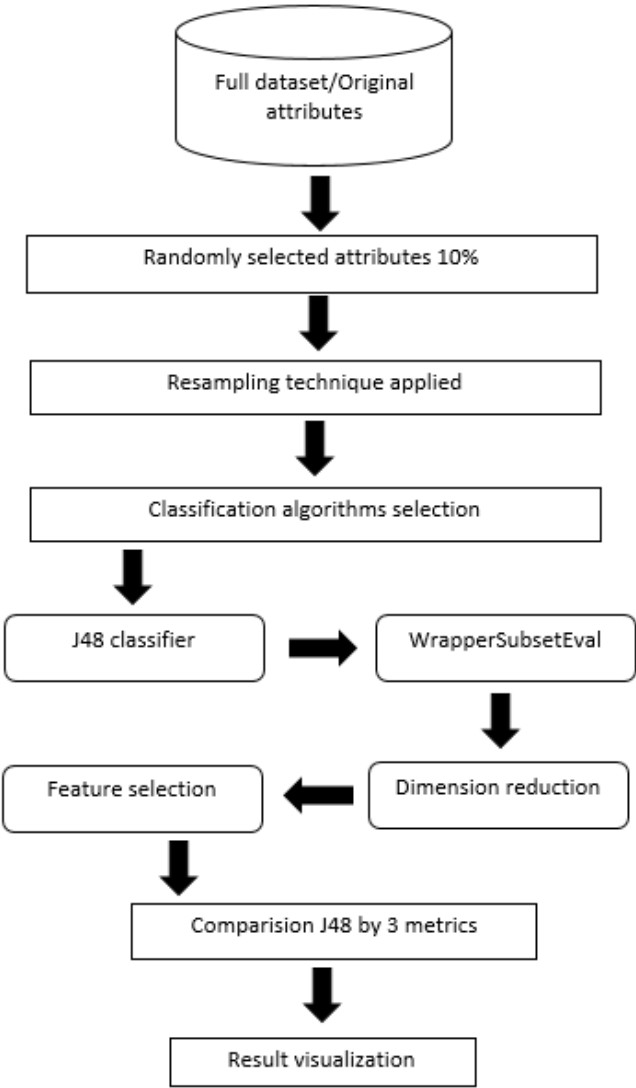

**Figure 1.** Methodological approach scheme.

*2.1. Bank Marketing Dataset Description*

The aim of this study focuses on selling long-term deposits by telemarketing. The result shows which is the better classifier for selling the deposit in the case of two algorithms with a binary outcome. This paper contains real data from a bank using 10% of the whole 41188 actual instances withtwenty-one identified attributesand two types of attribute ascategorical and numeric. The bank marketing dataset has been selected from the University of California at Irvine (UCI). They gathered the data for almost 2 years (2008 to 2010) from a Portuguese bank [11]. The dataset has 20 input variables and 1 output variable as a target. The characteristics of the dataset are shown in Table 1. They include both numerical and categorical variables, which can be fed into a classification algorithm to train a model in order to classify any new instances. As explained by [5], there are input variables characterizing the socio-demographics of the clients (e.g., job, marital status—in total 4 variables), the client-bank relationship (e.g., loans—in total 3), the contact context (e.g., previous campaign contact result—in total 8 variables), and the socio-economic macro context (e.g., Euribor interest rate—in total 5 variables).

**Table 1.** Dataset attributes description.

| Number | Dataset Attribute Description | Type | Detail |
|---|---|---|---|
| 1 | Age | Numeric | 18–88 |
| 2 | Job | Categorical | "management" "entrepreneur" "retired" "technician" "student" "blue-collar" "services" "admin." "self-employed" "unemployed" "housemaid" "unknown" |
| 3 | Marital status | Categorical | "single" "divorced" "married" "unknown" |
| 4 | Education | Categorical | "basic.9y" "high.school" "illiterate" "basic.6y" "professional.course" "basic.4y" "university.degree" "Unknown" |
| 5 | Default (has credit in default?) | Categorical | "no" "yes" "unknown" |
| 6 | Housing (has housing loan?) | Categorical | "no" "yes" "unknown" |
| 7 | Loan (has personal loan? related with the last contact of the current campaign) | Categorical | "no" "yes" "unknown" |
| 8 | Contact (contact communication type) | Categorical | "telephone" "cellular" |
| 9 | Month (last contact month of year) | Categorical | "Jul"–"Dec"–"Nov"–"Mar" "Aug"–"Oct"–"Apr"–"May" "Sep"–"Jun" |
| 10 | Day of week (last contact day of the week) | Categorical | "Monday" "Tuesday" "Friday" "Thursday" "Wednesday" |
| 11 | Duration (last contact duration, in seconds) | Numeric | 0–3643 |
| 12 | Campaign (number of contacts performed during this campaign and for this client) | Numeric | 42 different ones 1–56 |
| 13 | Pday (number of days that passed by after the client was last contacted from a previous campaign. 999 means client was not previously contacted) | Numeric | 27 different ones 0–999 |

**Table 1.** *Cont.*

| Number | Dataset Attribute Description | Type | Detail |
|---|---|---|---|
| 14 | Previous | Numeric | 0–7 |
| 15 | Poutcome (outcome of the previous marketing campaign) | Categorical | "failure" "nonexistent" "success" |
| 16 | Employment variation rate (quarterly indicator) | Numeric | −0.2, −2.9, −1.7, −1.8, −3, −0.1, 1.1, −1.1, −3.4, 1.4 |
| 17 | Consumer price index (monthly indicator) | Numeric | 92.201–94.767 |
| 18 | Consumer confidence index (monthly indicator) | Numeric | −26.9, −50.8 |
| 19 | Euribor3m (Euro Interbank Offer Rate 3 month) | Numeric | 0.634–5.045 |
| 20 | Number of employees (quarterly indicator) | Numeric | 4963.65228.1 |
| **Output variable** | | | |
| 21 | Customers term deposit (has the client subscribed a term deposit?) | Categorical | Yes–No |

### 2.2. Preprocessing

The original dataset consists of 41188 instances and 10% of the dataset was randomly selected. Therefore, there was a dataset of 4119 instances with one output variable and 20 input variables. There was no missing value in the dataset, but the dataset was imbalanced and only 451 samples were related to success, all other samples, 3668, belong to failure samples. Due to the large differences between the two samples, resampling techniques were applied to the dataset in order to address the problem of imbalanced data. The resampling technique is used to increase the accuracy of the forecast. Using the resampling technique helped to remove the imbalanced data and the difficulty that was identified in the dataset [15].

### 2.3. Data-Driven Models

There is a myriad of data-driven models to address classification problems. Some of the most widely adopted are grounded in statistical methods, such as logistic regression. Nevertheless, Artificial Intelligence (AI) has emerged to take advantage of the large amount of data available in today's world which is flooded with data. AI-based methods include evolutionary algorithms [16], modern optimization [17], genetic algorithms [18] and nature-inspired algorithms [19], among others. Such approaches can be applied to several different contexts, such as online learning, scheduling, multi-objective optimization, vehicle routing and medicine classifiers, among others, e.g., [20,21].

The complexity associated with AI-based approaches to data-driven problems and subsequent solutions still has an important drawback to overcome: since most managers do not have advanced algorithm backgrounds, they tend to be reluctant to adopt such approaches [4]. Typically, this occurs because AI methods are black-box, i.e., one cannot directly understand how the model makes a decision, whereas in comparison, one can easily understand an equation representing a logistic regression [22]. However, decision tree algorithms can be complex due to the training process that leads to the actual tree itself based on the Gini index coefficient (CART trees) or the entropy (C4.5) [23]. Thus, decision trees can usually achieve better performance than traditional logistic regression [5].

In this experiment, a decision tree classifier was used to improve the accuracy. Implementation was performed by a J48 data mining classifier to find the best results in terms of three important comparison metrics. As previously stated, the decision tree is a model that can be read and explained by humans.

The classification is due to the fact that the decision tree is being used to perform simple computations [24]. It is an appropriate model to predict nominal results that are the same as the prediction output of yes or no. It is one of the most famous and powerful classification algorithms of the decision tree [25]. J48 is an open-source java implementation of the C4.5 algorithm [23] used in this paper.

### 2.4. Feature Selection

Several introduced techniques of highly-related tasks, such as feature selection, are implemented in any data mining knowledge discovery projects, and were used to filter out irrelevant dataset features [21,26]. There are three groups of feature selection, they contain filters, wrappers and embedded methods [27]. In order to select an attribute, a dimension reduction method of a dataset is used, which can then discard the less important attributes. WrapperSubsetEval attribute evaluator was used to select the most useful attributes initially, then the J48 was used to remove unimportant attributes [28].

As a result, after loading the data onto the J48 classifier, due to the large and imbalanced dataset, a resampling technique was applied on the J48 model to modify the dataset and improve the accuracy [15]. Then the result, combined with feature selection to dimension reduction, selected the useful attributes for the J48 classifier. This method identified customer features, and removed the irrelevant data, and even reduced the extra features. Computational complexity was able to show the high output performance to help with decision making [29]. The combination makes the model faster, more efficient and even perform better. After all, the model result was compared against accuracy, sensitivity, and specificity metrics.

### 2.5. Evaluation Metrics

A confusion matrix shows the information about the correct and incorrect predictions with the knowledge of outcomes in Figure 2. Thus, a 100% accurate model will have 0 (zero) values for both false negatives (FN), i.e., the cases when the client actually subscribed to the deposit despite the prediction proposing that he/she wouldn't subscribe; and false positives (FP), i.e., the cases when the client did not subscribe to the deposit, although the model suggested that he/she would. Therefore, a confusion matrix is a useful tool for evaluating the system's performance [30].

**Figure 2.** Matrix confusion.

Based on the confusion matrix, it is possible to compute accuracy, sensitivity and specificity [31]. Accuracy, specificity, and sensitivity are the metrics that were used to evaluate the classification performance. Accuracy represents whole correctly classified numbers from the data and removes the incorrect numbers. How close the actual measured value is to the desired value is the main point of accuracy [32]. So, whenever the accuracy rate is higher, the result is better. Accuracy is not suitable enough for evaluating classification performance [33]. Therefore, most related metrics to the imbalanced data, namely sensitivity and specificity, were used in this study. Sensitivity and specificity are two metrics of statistics that are used to evaluate the results of binary (positive and negative) classification.

### 3. Results

The purpose of this study focuses on improving the sales of long-term deposits through telemarketing by forecasting the likely outcome of each contact. In this section, we show the results obtained by applying the classification techniques to the bank marketing dataset. Accuracy, sensitivity and specificity were computed to evaluate the performance of the J48 classifier in order to find any differences from the original data. Our results show that the best method for predicting the sale of the deposit was the decision tree algorithm (J48) with the binary result as the successful or unsuccessful outcome of each contact.

In the experiment, 10% (4119) of instances were randomly selected, and 21 attributes were used. A 10-fold cross-validation scheme was adopted to ensure independence between training and test sets [4]. In Table 2, we show the resulting confusion matrix. Based on these results, we also computed the accuracy, sensitivity and specificity (Figure 3 shows the values for J48).

**Table 2.** Confusion matrix for the three trained models.

| Classifier | TN | FN | FP | TP |
|:---:|:---:|:---:|:---:|:---:|
| J48 | 3535 | 133 | 243 | 208 |
| NB | 3331 | 337 | 273 | 278 |
| LR | 3570 | 98 | 254 | 197 |

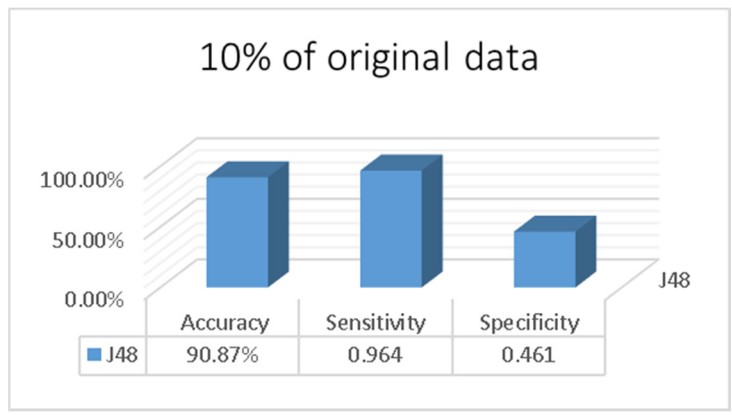

**Figure 3.** Accuracy, specificity and sensitivity for J48 decision tree.

After loading 10% of the original data, as discussed earlier, the dataset was imbalanced and only 451 samples corresponded to the success class, while 3668 samples failed. To solve the problem, a resampling technique was applied. Then, the success instances increased to 481 and the failure instances decreased to 3638. So, by changing the original data instances, after applying a resampling technique, we increased the correctly classified instances and reduced the incorrectly classified instances. Furthermore, the three metrics increment values, by this technique, are shown in Figure 4.

The main method used in this study was to combine a resampling technique with feature selection by using the J48 classifier. This method led to better results for accurately predicting customer behavior based on the account of long-term deposits.

WappedSubsetEval was used to select the useful features and remove the other extra ones; it provided dimension reduction in order to improve speed, accuracy, efficiency and performance. This method helped to obtain better results through the J48 classifier, as shown by the three metrics. Resampling and feature selection have a direct efficacy impact on the accuracy rate of classifier algorithms, as highlighted by [34].

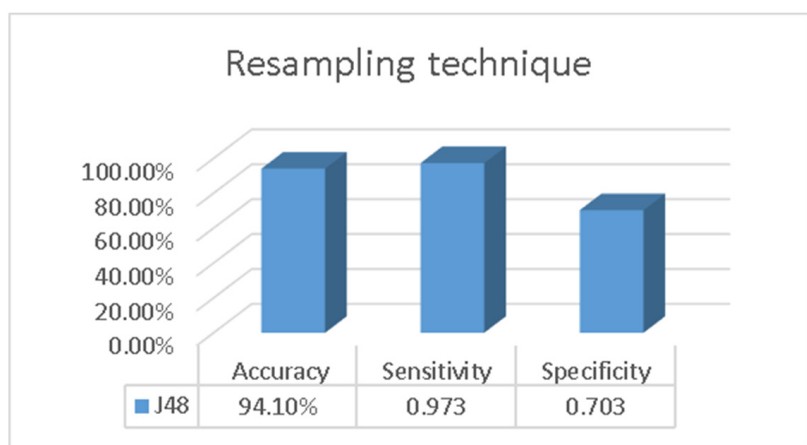

**Figure 4.** Compared accuracy, specificity and sensitivity for J48 after using a resampling technique.

When using the attribute evaluator for the J48 model, the 12 selected attributes from the 20 that were input, resulted in an increased number of correctly classified instances, as shown in Table 3.

**Table 3.** Variables selected after using WappedSubsetEval to dimension reduction.

| J48 | |
|---|---|
| 1 | Age |
| 2 | Education |
| 3 | Contact |
| 4 | Duration |
| 5 | P days |
| 6 | Previous |
| 7 | P outcome |
| 8 | Emp.Var.Rate |
| 9 | Cons.Price.Idx |
| 10 | Cons.Conf.Idx |
| 11 | Euribor3m |
| 12 | Nr.Employed |
| 13 | Y as output |

Feature (or variable) selection keeps only the useful features, resulting in better classification and also provides a tuned binary tree from the J48 model [29]. The model achieved an accuracy of 94.39%, a sensitivity of 0.975 and 0.709 specificity. The result of the method showed an improvement of customer prediction accuracy (Figure 5).

For both techniques, the confusion matrix shows information about the instances classified as false positive (FP), false negative (FN), true positive (TP) and true negative (TN) that are shown in Figure 6. The values for Table 2 are exactly the same as Figure 6. The comparison of the three levels that are used clearly shows the benefit of using both resampling and feature selection, with higher values of TP and TN.

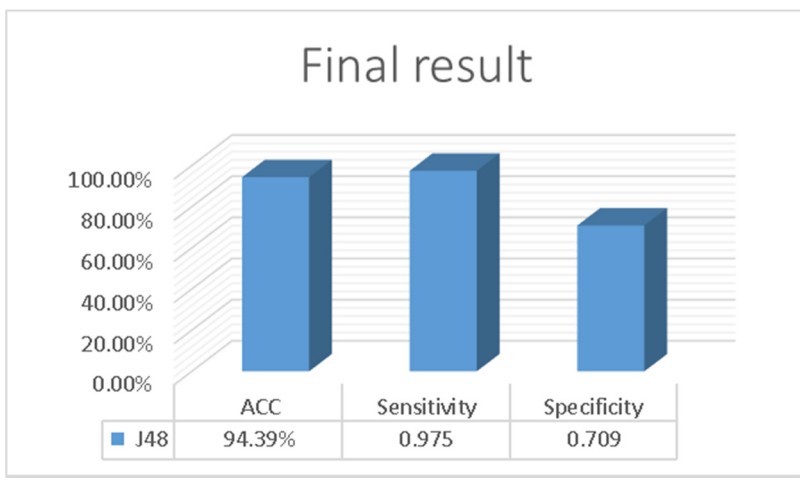

**Figure 5.** Compared models and methods after the last process.

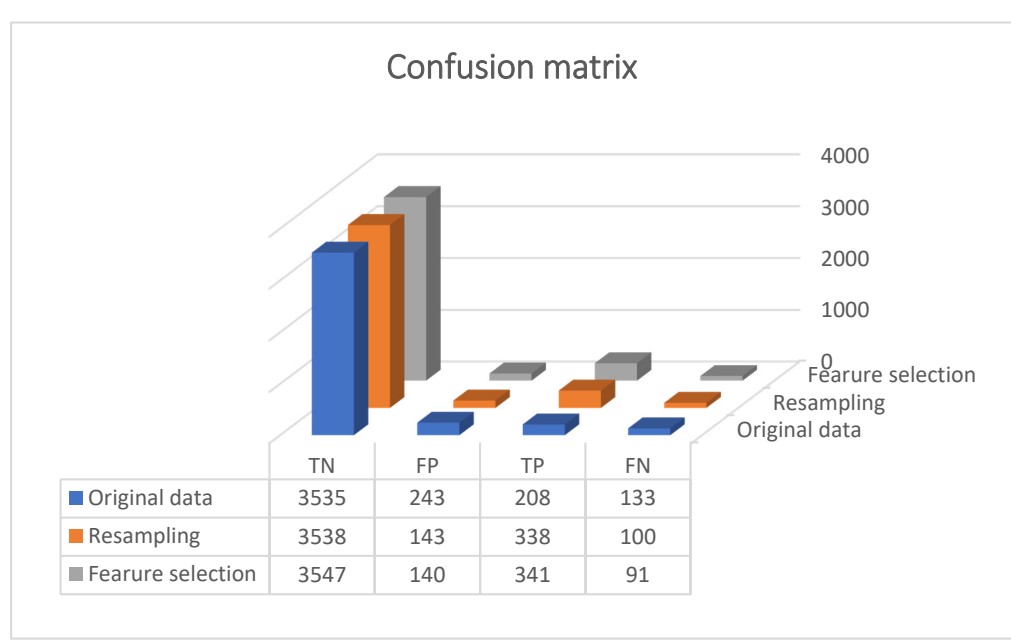

**Figure 6.** Shows matrix confusion at three levels when comparing the results.

## 4. Discussion

The real data used in this paper was the full bank marketing of 41188 instances in UCI from a Portuguese bank [11]. The main goal of this study was to understand the effectiveness of the J48 model at predicting the success of telemarketing calls for selling bank long-term deposits. We showed how the J48 classifier could be improved to achieve better evaluation metric results for the three used metrics. The metrics used for measuring the performance of the classifier were accuracy, specificity and sensitivity. Several studies have reported using J48 for bank direct marketing. After comparison, we proved that in this study the J48 performs better when using resampling and feature selection for the three metrics, with a result of 94.39% for accuracy, 0.975 for sensitivity and 0.709 for specificity.

In today's competitive market, companies must make critical decisions to affect their future. That is the most common mistake made that causes the wrong marketing approaches. All managers wish to have faithful customers and increase revenue to attract customers, maintaining their presence is something else. Retaining customers and customer loyalty is more important in a competitive business where it is hard to stand out. In order to have customers who are satisfied and to solve the problem of customer loyalty and retention, the J48 classifier is suggested in order to predict the sale of long-term deposits.

One of the most common problems of telemarketing is that customers may feel annoyed if they are not really interested in the product being offered.

This paper achieved better results in comparison to other previous studies. In our preliminary study [35], five DM models: Naive Bayes (NB), logistic regression (LR), decision tree (DT), multilayer perceptron neural network (MLPNN) and support vector machine (SVM) were compared to predict the success of selling long-term deposits by telemarketing calls in bank marketing. LR has achieved the better performance than others based on accuracy: 91.21%, ROC 0.93 and sensitivity, 0.912%, from the same dataset [11].

Ref. [36] used a set of training data and tested it with C4.5 and Naive Bayes algorithms in a group of bank customers. Then, they were able to improve the accuracy results by about 1%. They obtained 93.961% for C4.5 and 87.724% for Naive Bayes. Ref. [37] cleaned and prepared the bank dataset [11] by considering data integrity, unified types, missing value imputation, data transformation, and data balancing, then applied five machine learning algorithms. Finally, they achieved the best accuracy by using the Gaussian NB algorithm, 88.86%. Ref. [38] proposed a geometric data perturbation (GDP) method using data partitioning and three-dimensional rotations. They divided attributes into three groups so that each group rotated into different axis to provide privacy, increased accuracy and efficiency. They used DT-IBK-J48-NB to improve the accuracy of the J48 with the three-dimensional rotation transformation (3DRT) technique which is four times better than the 2DRT. Ref. [39] analyzed bank customer information for predicting the success of bank telemarketing. They proposed the prediction model FMLP-SVM method. This model is using a fuzzy algorithm, it first fuzzifies input variables and uses fuzzified MLPNN as a function of the fuzzy SVM kernel function, then uses fuzzified data to predict the bank telemarketing. They used four algorithms (DT, SVM, Bayes net, FMLP-SVM) to increase the accurate result so as to increase the overall accuracy. Then, FLIP-SVM could gain the most accurate result at an accuracy level of 92.89% and DT could have an accuracy of 89.66%, with a 20% sample size. Another challenge of data mining is handling the problem of imbalanced data. A dataset is imbalanced when one class dataset has much fewer numbers of instances than the other class dataset [40]. Ref. [15] proposed a model based on a SMOTE algorithm to handle imbalanced datasets. SMOTE is an oversampling technique and it was applied on the minority class in order to create an increment of the data. After the SMOTE method was applied on the imbalanced dataset, an increment in the minority class was accompanied by no change in the majority class. The results showed that, the SMOTE method can solve the imbalanced data problem. The results from comparing the proposed methods with J48, rotation forest, Bayes net, Naive Bayes, MLP and RBF neural network were presented, and J48 achieved 89.43% accuracy on the bank direct marketing dataset [11]. The achieved metrics are compared with the ones obtained by other studies in Table 4.

In summary, the contribution of this research is as follows: we addressed the imbalanced issue of the dataset and improved prediction results after selecting 10% of the full dataset [11]. A resampling technique was used to handle this problem and complexity of data computing.

After using a supervised resampling technique, the minor class increased (over sampling) and the major class decreased (under sampling) at the same time. This technique was effective for the J48 classifier of the bank direct marketing imbalanced dataset [11] to more accurately predict the resample technique, combined with feature selection, and applied to find the minimum subset of features that captures the related attributes of a dataset in order to enable sufficient classification. Feature selection is a technique used to remove the irrelevant features that are not practical for making the learning algorithm operate faster and more efficiently [29]. We identified the most promising predictive input: 12 features out of the 20 input from J48 model. As a result, the dimension reduction increased the effectiveness of the three metrics that we used in this paper. Then, we could further improve the accuracy rate and even the highest speed of prediction. Finally, we highlight that, from a managerial perspective, our best approach using J48 with resampling

and feature selection resulted in the lowest values of false positives and false negatives. This is especially important in the telemarketing business, as false positives represent cases in which calls are made (consuming both time and money) needlessly, since the contacted customers did not subscribe to the product, whereas false negatives are even worse for marketing managers, since these represent cases where the model suggested that the customers would not subscribe to the deposit, which was actually not true, and implies the loss of business opportunities.

**Table 4.** Metrics comparison with previous studies.

| Categories | References | Methods | Algorithm | Best Accuracy |
|---|---|---|---|---|
| | Current paper | Resampling and feature selection | J48 | 94.39% |
| | [35] | Training and tested | C4.5 | 93.96% |
| | [36] | Compared 5 algorithms | LR | 91.21% |
| Bank dataset [11] | [37] | Data preparation by preprocessing | NB | 88.86%. |
| | [38] | Data partitioning and three-dimensional | J48 | <89% |
| | [39] | Using fuzzy algorithm model | FMLP-SVM | 92.89% |
| | [40] | Using oversampling (SMOT) | J48 | 89.43% |

## 5. Conclusions

This research resulted in a classifier to predict whether a customer will subscribe to a long-term deposit or not. The combination of resampling and feature selection has been applied to 10% of the whole dataset (41188), resulting in 4119 instances from the UCI repository of a Portuguese bank. The proposed method addresses the imbalance issue and also removes unnecessary features through dimension reduction, based on the J48 model. To evaluate the performance of the classifier, three metrics, specificity, sensitivity and accuracy, were calculated. A comparison of the metrics could prove that the attribute evaluator has increased the important values of accuracy (94.39%), sensitivity (0.975) and specificity (0.709) by the J48 classifier. The evaluator selected 12 attributes out of the 20 attributes that were input according to the output. The aim of this paper was to predict the success of selling long-term deposits by telemarketing.

Nevertheless, this study has important limitations that need to be addressed, which can lead to new future research directions, as follows:

- This paper was limited to the telemarketing case—in the future, we aim to assess our approach on other marketing cases and datasets, such as direct marketing through email or instant messaging.
- While we have shown that resampling and feature selection improved the J48 algorithm beyond the results achieved by previous studies, we still do not have evidence if such an approach (resampling plus feature selection) would significantly improve other algorithms such as those based on neural networks, perhaps with results that are better than those from the J48 overall.

**Author Contributions:** Conceptualization, F.S.; methodology, F.S.; software, F.S.; validation, F.S. and S.M.; formal analysis, F.S.; investigation, F.S.; resources, F.S.; data curation, F.S.; writing—original draft preparation, F.S.; writing—review and editing, S.M., and F.S.; visualization, F.S.; supervision, S.M.; funding acquisition, S.M. Both authors have read and agreed to the published version of the manuscript.

**Funding:** This work is funded by national funds through FCT—Fundação para a Ciência e Tecnologia, I.P., under the project FCT UIDB/04466/2020.

**Institutional Review Board Statement:** Not applicable.

**Informed Consent Statement:** Not applicable.

**Data Availability Statement:** The adopted dataset can be obtained from https://archive.ics.uci.edu/ml/datasets/Bank+Marketing (accessed on 22 July 2021).

**Conflicts of Interest:** The authors declare no conflict of interest.

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
