# Peer review of "Improving the Accuracy of Predicting Bank Depositor’s Behavior Using a Decision Tree"

_applsci, doi:10.3390/app11199016_

Round 1

Reviewer 1 Report

This study focuses on improving the accuracy of predicting bank depositor' behavior using decision tree. I think the paper fits well the scope of the journal and addresses an important subject. However, a number of revisions are required before the paper can be considered for publication. There are some weak points that have to be strengthened. Below please find more specific comments:

*The abstract seems kind of short. I suggest adding one or two sentences highlighting the outcomes of this work and contributions to the state-of-the-art.

*The literature review section seems to be kind of short as well (partially discussed in the introduction section). Please check more recent studies to make sure that the most recent and relevant studies are captured.

*Page 3: It would be good to add the fourth column in Table 1 with some ranges (i.e., what are the age ranges for the considered dataset, how many job categories were identified and what are they, etc.).

*Page 3: The authors discuss the adopted data mining models in section 2.3. Before describing the adopted data mining models, the authors should create a discussion highlighting a wide implementation of different AI methods (e.g., heuristics, metaheuristics, nature-inspired algorithms) – not just data mining models. There are many different domains where various AI methods have been applied as solution approaches, such as online learning, scheduling, multi-objective optimization, vehicle routing, medicine, data classification, and others (not just prediction of the bank depositor' behavior). The authors should create a discussion that highlights the effectiveness of different AI methods in the aforementioned domains. This discussion should be supported by the relevant references, including the following:

  • Zhao, H. and Zhang, C., 2020. An online-learning-based evolutionary many-objective algorithm. Information Sciences, 509, pp.1-21.
  • Dulebenets, M.A., 2021. An Adaptive Polyploid Memetic Algorithm for scheduling trucks at a cross-docking terminal. Information Sciences, 565, pp.390-421.
  • Liu, Z.Z., Wang, Y. and Huang, P.Q., 2020. AnD: A many-objective evolutionary algorithm with angle-based selection and shift-based density estimation. Information Sciences, 509, pp.400-419.
  • Pasha, J., Dulebenets, M.A., Kavoosi, M., Abioye, O.F., Wang, H. and Guo, W., 2020. An Optimization Model and Solution Algorithms for the Vehicle Routing Problem with a “Factory-in-a-Box”. IEEE Access, 8, pp.134743-134763.
  • D’Angelo, G., Pilla, R., Tascini, C. and Rampone, S., 2019. A proposal for distinguishing between bacterial and viral meningitis using genetic programming and decision trees. Soft Computing, 23(22), pp.11775-11791.
  • Panda, N. and Majhi, S.K., 2020. How effective is the salp swarm algorithm in data classification. In Computational Intelligence in Pattern Recognition (pp. 579-588). Springer, Singapore.

Such a discussion will help improving section 2 significantly. After this discussion it would be logical to specifically describe the adopted data mining models.

*Please provide a bit more details regarding the input data used for numerical experiments.

*The results from numerical experiments could be presented in a more comprehensive manner. I suggest adding more supporting discussions.

*Page 10: The conclusions section should expand on limitations of this study and future research needs. I suggest listing the bullet points.

Reviewer 2 Report

The first part of the document needs moderate english changes, there are problems of stye and wording. The work is not properly documented. It seems a work well designed but with low documentation, for instance the variables, what variables were considered? what is Default, Previous, Poutcome, Peuibor3m? most are meaningless. A confusion matrix is used; so what? no documentation on the tools and from the brief explanations until section 3 it is mostly results. The way it is presented is like; we analized "things" and with several approaches we improved the results. Again, I don't question the work but a publication either is a self-report on "things" or the reader finds it relevant in scientific terms. Then how can someone find useful this work if there is no documentation on the variables used or excluded. If the authors want, it could become an interesting document by documenting better what is reported.

Round 2

Reviewer 1 Report

The authors took seriously my previous comments and made the required revisions in the manuscript. The quality and presentation of the manuscript have been improved. Therefore, I recommend acceptance.